# Top-down patterning of topological surface and edge states using a focused ion beam

Abdulhakim Bake [1,2,11], Qi Zhang [2,3,11], Cong Son Ho [4], Grace L. Causer [5], Weiyao Zhao [1,2], Zengji Yue [6,7], Alexander Nguyen[2,8], Golrokh Akhgar[2,8], Julie Karel[2,8], David Mitchell [9], Zeljko Pastuovic[10], Roger Lewis[7], Jared H. Cole [2,4], Mitchell Nancarrow[9], Nagarajan Valanoor [2,3], Xiaolin Wang[1,2] & David Cortie [1,2,10] ✉

The conducting boundary states of topological insulators appear at an interface where the characteristic invariant $\mathbb{Z}_2$ switches from 1 to 0. These states offer prospects for quantum electronics; however, a method is needed to spatially-control $\mathbb{Z}_2$ to pattern conducting channels. It is shown that modifying $Sb_2Te_3$ single-crystal surfaces with an ion beam switches the topological insulator into an amorphous state exhibiting negligible bulk and surface conductivity. This is attributed to a transition from $\mathbb{Z}_2 = 1 \rightarrow \mathbb{Z}_2 = 0$ at a threshold disorder strength. This observation is supported by density functional theory and model Hamiltonian calculations. Here we show that this ion-beam treatment allows for inverse lithography to pattern arrays of topological surfaces, edges and corners which are the building blocks of topological electronics.

The sub-classes of quantum insulators can be distinguished using invariants (e.g., $\mathbb{Z}_2$)—simple groups of integers related to the Berry curvature—which also encode information on their distinctive physical properties[1–8]. A special case is the 3D strong topological insulator (TI) having one "strong" index $\mathbb{Z}_{2s} = 1^{*}$. The non-zero strong index is the criterion for the existence of zero-gap spin-helical Dirac surface states which appear within the bulk electronic bandgap, as observed experimentally[9–17]. These states offer the prospect of dissipationless transport for nanoelectronics, enabling topological transistors to potentially evade the so-called "Boltzmann's tyranny" of conventional semiconductors[18], together with enticing prospects for topological qubits based on Majorana zero modes[19]. Although dissipationless channels have already been observed at the natural edges of TI crystals[20], a missing step is a precise top-down method to spatially engineer nanoscale arrays of conducting channels for scalable integrated circuitry, which would require regions of both $\mathbb{Z}_{2s} = 1$ and $\mathbb{Z}_{2s} = 0$. Adopting the surface engineering paradigm from silicon technology, an effective way to do this would be to define the channels using lithography of monolithic crystal surfaces enabled by ion beams[21–23]. So far there has been no demonstration of this technique, which is well-known in other microelectronics, to pattern the position of topological edge states. Here we demonstrate the ability to perform top-down nanopatterning of topological surface and edge states at the $Sb_2Te_3$ surface using an ion beam to induce atomic displacements that pin the edge states at the boundary with local amorphous regions. We find that, with ion beam processing, we can study and control three types of interfaces in a topological insulator, each with distinctive electronic properties. The nomenclature used here to describe these three interfaces is A–V for amorphous–vacuum, C–V for crystalline–vacuum, and A–C for amorphous–crystal boundaries.

[1]Institute for Superconducting and Electronic Materials (ISEM), University of Wollongong, Wollongong, NSW 2522, Australia. [2]The Australian Research Council Centre for Excellence in Future Low Energy Electronics Technologies, Wollongong, Sydney, Melbourne, Australia. [3]School of Materials Science and Engineering, The University of New South Wales, Kensington, NSW 2052, Australia. [4]Chemical and Quantum Physics, School of Science, RMIT University, Melbourne, Australia. [5]Physics Department, Technical University of Munich, 85748 Garching, Germany. [6]Institute of Photonic Chips, University of Shanghai for Science and Technology, Shanghai 200093, China. [7]School of Physics, Faculty of Engineering and Information Science, University of Wollongong , Wollongong 2522 NSW, Australia. [8]Department of Materials Science and Engineering, Monash University, Clayton, VIC 3800, Australia. [9]Electron Microscopy Centre, University of Wollongong, Wollongong, NSW 2522, Australia. [10]The Australian Nuclear Science and Technology Organisation (ANSTO), Lucas Heights, NSW 2234, Australia. [11]These authors contributed equally: Abdulhakim Bake, Qi Zhang. ✉e-mail: dcortie@uow.edu.au

Furthermore, we observe modified conductivity from 2D topological "surface states" and 1D "edge" regions.

Determining if amorphous "glassy" states at an A-V 2D surface are topological or trivial is currently of great interest[24–26], and here we first focus on this question experimentally. This question is the main unknown, and is of key importance to surface engineering, as it determines whether disordered regions behave as a "topological vacuum". Initial theories predicted that, while the TI Dirac band-structure is resilient against low energy charge disorder, strong disorder from lattice defects can introduce an electronic gap and transform the amorphous system into a trivial Anderson insulator, with $\mathbb{Z}_{2s} = 0$ (i.e. equivalent to that of vacuum)[27]. More recent theories, in contrast, predict that the collapse of the topological state in the glass is not guaranteed, and instead, solid-state amorphous topological insulators do exist in special cases, at least in numerical simulations[25,28,29], and model photonic/mechanical metastructures[30–32]. This is expected to be non-universal as the emergence of topological edge states requires very special interactions in the semi-random glass, and for example, Voronoi-type amorphous networks of interconnected points are predicted to be topological insulators, whereas those formed from triangulation are not[25]. In practice, it is unclear which conditions apply in real solids. Recent theories are divided on this point for the prototypical chalcogenide family of TIs[33,34] where experimental information on the glassy state is also limited[35,36]. This motivated a detailed systematic experimental study of the electronic properties of $Sb_2Te_3$ as it transforms from the crystalline state into a glass.

Crystalline $Sb_2Te_3$ is a textbook example of a $\mathbb{Z}_2$ 3D TI described by the space-group (SG) $R\bar{3}c$ #166 (illustrated in Fig. 1A), which possesses inversion symmetry. As a good example of a strong TI, as designated in the topological materials database (TMD)[15], it hosts a single Dirac cone in its surface electronic band structure, with a bulk-band gap of 0.2–0.3 eV, as detected by angle-resolved photoemission spectroscopy (ARPES) and magneto-transport experiments[9–17]. However, the related high-pressure $Sb_2Te_3$ compound of the C2/m space group (SG #12, Fig. 1B) has disordered Sb/Te sites and is completely topologically trivial with $\mathbb{Z}_{2s} = 0$ and $\mathbb{Z}_{ws} = 0$ (TMD #187539), therefore having an absence of topologically protected surface states. The topology of amorphous $Sb_2Te_3$ (Fig. 1C) is unknown, however, in this work, we will show that being structurally similar to the C2/m phase, it behaves as a trivial insulator, and hosts no detectable surface states. This intuition is supported by the past literature, where the insulating properties of amorphous $Sb_2Te_3$, both in the bulk and surface, have been established as a key property in its role as a phase change memory (PCM) material[37–39]. The collapse of the topological state has also recently been inferred in disordered $Sb_2Te_3$ thin films[34,36], although the growth-induced disorder makes it difficult to achieve top-down patterning in that case.

## Results

To parametrize the level of surface disorder experimentally, $Sb_2Te_3$ single crystals were grown and polished and the surface exfoliated to present (001) planes with flat regions interspaced by terraces. The smooth regions of the crystalline surfaces were then exposed to a focused-ion beam (FIB) at low energy (8–30 keV) using a systematic set of ion beam fluences ($1 \times 10^{11}$ to $5 \times 10^{16}$ ions per cm²) (see Supplementary Figs. 1–15). The initial crystals are typical of $Sb_2Te_3$, showing well-defined Shubnikov–de Haas oscillations, a non-trivial Berry phase, high mobility, and residual bulk conductivity, as reported previously[40] (see also Supplementary Fig. 16). The incident ions modify the first few nanometers of the surface, as shown schematically in Fig. 1D, primarily by displacing Sb and Te from their chemically ordered crystalline sites, (see Monte Carlo simulations in Supplementary Fig. 2), in turn modifying the electronic structure. Low-energy gallium ion beams were found to be effective for modifying the surface, as they cause a large amount of lattice displacement, with a sharply defined stopping depth, and do not lead to porosity or flaking. The small residual $Ga^{3+}$ impurity is also isovalent with the $Sb^{3+}$, minimizing electronic doping.

Using this method, the $Sb_2Te_3$ surface can be tailored with high precision so that the top 20 nanometers of the irradiated surface where the ion beam is focused becomes amorphous, as shown by the cross-sectional scanning transmission electron microscope (STEM) image (Fig. 1E), whereas unirradiated regions remain crystalline. The transition from the crystalline to the amorphous phase occurs over a small spatial scale of 2–4 nm (Fig. 1F). Lateral patterns can be formed

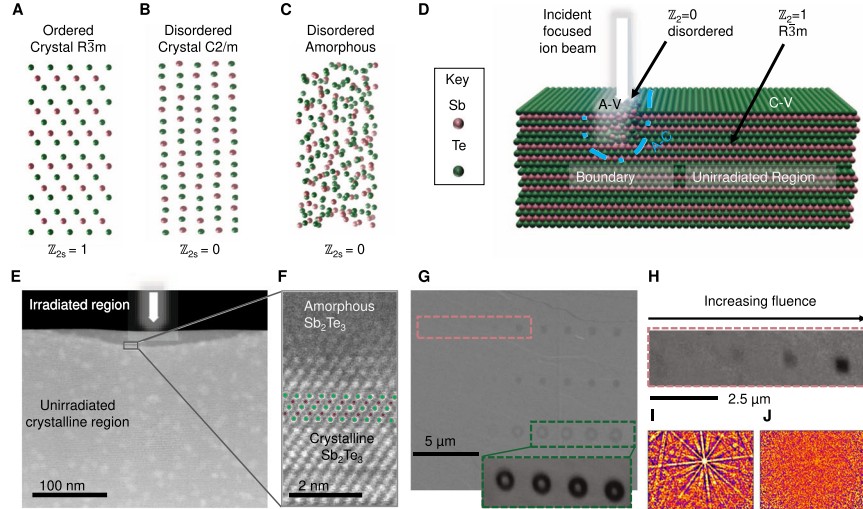

**Fig. 1 | The crystalline state of the topological insulator $Sb_2Te_3$ can be modified using a focused-ion beam. A** Illustration of the ordered crystalline $R\bar{3}m$ phase of $Sb_2Te_3$ which has $\mathbb{Z}_{2s} = 1$. **B** The disordered C2/m phase has random Sb/Te sites and has $\mathbb{Z}_{2s} = 0$. **C** Amorphous $Sb_2Te_3$ shares similarities with the disordered crystalline phase. **D** Schematic illustration of the ion-beam assisted amorphization process. **E** Cross-sectional scanning transmission electron image in HAADF mode of the irradiated region at the surface of the crystal. **F** High-resolution HAADF image showing the amorphous and crystalline regions where the irradiation used was $2.2 \times 10^{15}$ ions/cm² at 8 keV. **G** Secondary electron image of a plane view of the surface showing the ion-beam irradiated patterns (inset image shows the EBSD band contrast image of a selected region). **H** EBSD band contrast image from a region in (**G**) (pink dashed region). Electron backscattered diffraction patterns (**I**) from the lightly irradiated pattern and (**J**) from the highly irradiated pattern in (**H**), showing the transition from order to disorder as the ion fluence is increased (see Supplementary Table 2 and Supplementary Fig. 7).

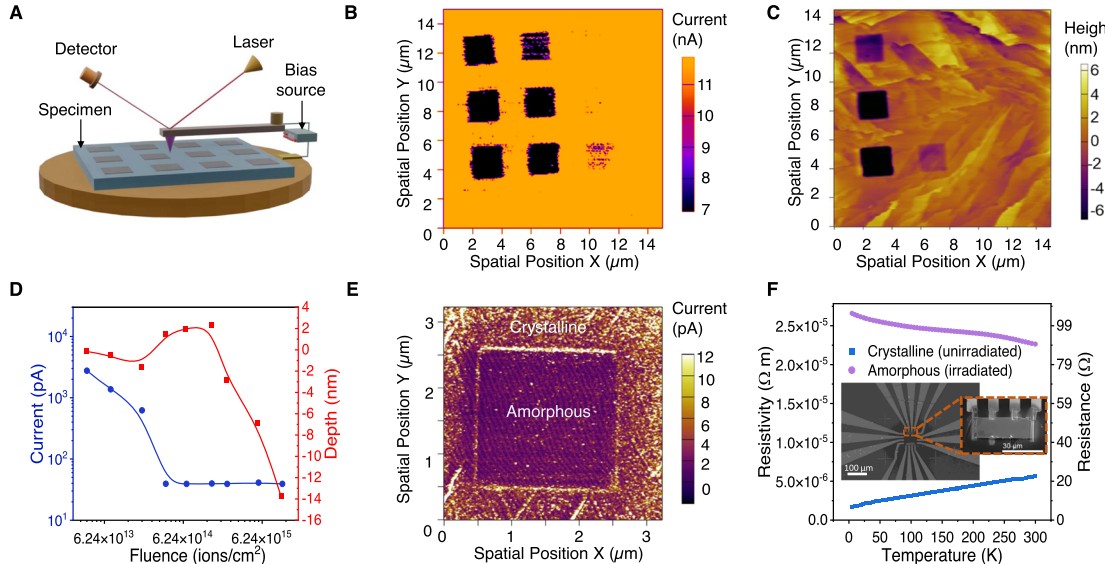

**Fig. 2 | The disordered regions modified by the ion beam have suppressed surface conductivity. A** Schematic of the AFM configuration showing how current and topography is measured over a patterned region. **B** Mapping of surface currents of a $Sb_2Te_3$ single crystal showing that the $3 \times 3$ rectangular grid sections, which were modified by the ion beam, are less conducting. Each square received a sequentially increasing ion beam fluence from low dose (top right square) to high dose (bottom left square) ranging from $3.74 \times 10^{13}$ ions/cm² to $1.12 \times 10^{16}$ ions/cm² (see Supplementary Information for the details) The measurement bias was 2.5 V. **C** Height map over the same region as in (**B**) showing that topography is modified for higher fluences. **D** Integrated current and height of each $Sb_2Te_3$ grid point irradiated with increasing dose of $Ga^+$ ions, measured using cAFM with a scanning bias of 2.2 V. **E** Current mapping using a 1 V DC bias for an individual irradiated box (see Supplementary Information for IV characterization) showing additional conductivity at the edges of the pattern, and on surface terraces. **F** Four-probe temperature-dependent resistance of a $Sb_2Te_3$ device before and after amorphization, showing the increased resistance. The inset shows the measurement configuration.

by raster scanning the beam across the surface, or using a mask, and the resulting modification depends on the ion fluence. A plane view scanning electron microscopy (SEM) image is shown of a surface irradiated to form various geometric patterns with fluences systematically ranging from $1 \times 10^{12}$ to $1 \times 10^{15}$ ions/cm² (Fig. 1G). Three distinct regimes are observed. At low fluences (left), there is no detectable change in the surface. At high fluences (right), the ion beam ablates the surface, resulting in microscale trenches (e.g. a physical vacuum). The most interesting situation occurs at intermediate fluences where there is a clear change in electron contrast, but the surface is flat down to the nanoscale (see atomic force microscopy image later). In this regime, the ion irradiation introduces lattice disorder and mixes atomic Sb/Te sites leading to a situation similar to Fig. 1B, C. The change in the atomic structure leads to modified contrast in the electron backscattering diffraction (EBSD) maps (Fig. 1H), which probe the surface crystal structure from the exit beam Kikuchi diffraction patterns within the first few nanometers near the surface[41–43]. The irradiated regions become amorphous as shown by the lack of Kikuchi patterns, whereas the unirradiated surface is crystalline (Fig. 1I, J). The pattern for the unirradiated points can be indexed to the crystalline unit cell of $R\bar{3}c$ ordered $Sb_2Te_3$ (the Supplementary Information contains a full EBDS band map, Euler orientation, and elemental map). In contrast, the highly irradiated region shows a featureless pattern (Fig. 1J) characteristic of a non-crystalline structure. The disappearance of Kikuchi bands, together with the loss of lattice fringes in the STEM imaging (Fig. 1E) shows the system transitions to the amorphous state only where it has been impacted by the ion beam irradiation. This demonstrates that it is possible to precisely engineer an amorphous phase transition in a local surface region whilst preserving the surrounding crystallinity which is a prerequisite for well-controlled lithographic patterning.

The ion-beam patterned regions show strongly modified electronic properties, as detected using a conductive atomic force microscopy (cAFM) and standard transport measurements (see below). cAFM has the advantage of mapping the local conductivity,

which can be also correlated with the physical topography (height). For this purpose, a grid of squares was patterned on the $Sb_2Te_3$ using fluences near the amorphization threshold (Fig. 2A). The fluence applied to each of the grid points was varied systematically. The squares irradiated with higher fluences show significantly lower conductivity (Fig. 2B) and the observed currents are vanishingly small (pA, at the instrumentation limit). These insulating regions appear dark on the current map. In contrast, unirradiated or lightly irradiated regions (fluence $<6.24 \times 10^{13}$ ions/cm) exhibit currents which are 2–3 orders of magnitude larger (nA) for the same scanning bias (1–2.2 V), indicating a highly conductive surface. The current map shows a clear step function in behavior with Ga ion fluence. Grids irradiated with fluences above $1.87 \times 10^{14}$ ions/cm² (8 keV incident energy) show a dramatic decrease in conductivity. This step function behavior is attributed to the point where the amorphous layer, observed by the EBSD and STEM in Fig. 1, becomes sufficiently thick to disrupt the electronic transport through the surface regions. Importantly, the remarkably large change in the surface conductivity occurs in the room temperature data in Fig. 2F which implies a radical change in the electronic structure, rather than a minor doping effect. As a consequence of the cAFM measurement geometry, any observed current must flow both through the bulk and surface regions of the crystal, and both in-plane and out-of-plane currents may contribute, depending on the position of the tip relative to the pattern. We, therefore, conclude that both the surface and the bulk of the amorphous $Sb_2Te_3$ irradiated regions become highly insulating, consistent with the collapse of the topological bulk state into a trivial insulator. To provide complementary insights into the electronic properties, simple four-probe devices were made by extracting a small and thin (640 nm-thick) rectangular region from the crystalline $Sb_2Te_3$ surface using a nanomanipulator. A custom FIB-lift out procedure was developed to transfer crystalline $Sb_2Te_3$ chips to a premade set of contacts (inset in Fig. 2F) whilst preserving the crystallinity of the surfaces (i.e., the top and bottom surfaces are virgin, unexposed to an ion beam). At these thicknesses, the surface-region transport is expected to account for a substantial fraction of the

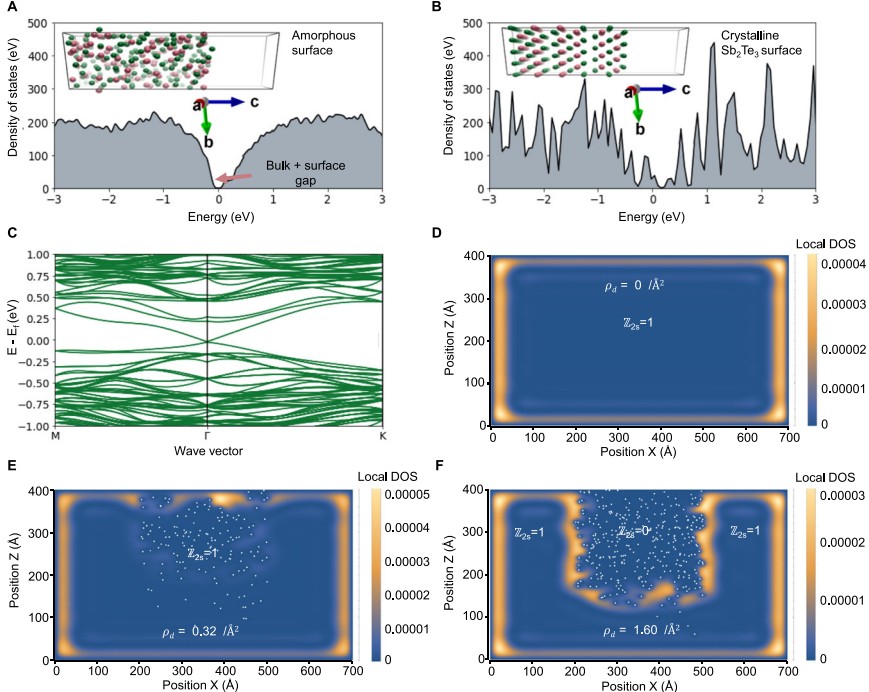

**Fig. 3 | Theoretical calculations show that while crystalline Sb₂Te₃ hosts Dirac surface states, the amorphous version does not, consistent with the $\mathbb{Z}_2 = 1 \rightarrow \mathbb{Z}_2 = 0$ transition. A** The electronic density of states of an amorphous Sb₂Te₃ surface exhibits a bandgap with no states crossing the Fermi level. The inset shows the atomic model of the amorphous Sb₂Te₃ surface slab generated from the molecular dynamics quench, used to calculate the electronic structure with a 15 Å vacuum boundary in the $c$-direction. **B** The electronic density of states of the crystalline Sb₂Te₃ surface has states crossing the Fermi level. The inset shows the crystalline surface slab used to calculate the electronic structure. **C** The band structure of the crystalline surface shows the characteristic Dirac surface states crossing the Fermi level. **D** The real-space image of the defect-free crystal shows the electron density calculated with the model Hamiltonian and exhibits surface states at the crystal-vacuum interface. **E** A moderate level of defects does not remove the surface state position. Each white dot represents a defect in the system. **F** At a critical threshold of disorder, the disordered region becomes non-topological ($\mathbb{Z}_2 = 0$) as the surface states then reposition to wrap around the new topological vacuum region.

overall conductivity. The chip was transferred to a cryostat and measurements of resistivity shows the nearly metallic behavior commonly found in crystalline Sb₂Te₃ (Fig. 2F). The chip was subsequently transferred back to the focused ion beam mill and the entire top surface was exposed to a high Ga dose, before remeasuring the resistance. The same device after ion beam amorphization had a resistance that was two orders of magnitude higher than in the unirradiated state at 3 K. Moreover, in contrast to crystalline Sb₂Te₃, the resistance of amorphous Sb₂Te₃ increased with decreasing temperature. This is a signature of a trivial insulator ($\mathbb{Z}_2 = 0$) since topological states would be expected to introduce a resistivity plateau. As the FIB devices have a parallel bulk conduction channel from the residual crystalline region (see Supplementary Information Section 4.5), we also reproduced these findings for MBE-grown ultra-thin Sb₂Te₃ films on insulating alumina finding that, in the absence of the bulk channel there is an even more impressive increase in resistance (Supplementary Fig. 19). Similar high-resistance states have been reported in the literature for Sb₂Te₃ thin films in phase change memory devices[37–39]. Together with the cAFM observation, the direct transport measurements strongly suggest the irradiated amorphous regions at the A-V boundaries become trivial insulators and therefore form a topological vacuum ($\mathbb{Z}_2 = 0$)[44].

The experimental evidence for the insulating 2D A-V boundaries is extremely robust, and it is also supported by the density functional theory (described in a later section) which explains this on the basis of a topological transition to $\mathbb{Z}_2 = 0$. The observation of the highly insulating A-V 2D region is the main claim of our work as it appears consistently in all patterned regions above the critical fluence level. There are, however, also interesting secondary effects that occur at many of the 1D A-C boundaries which warrant discussion. Consistent with the

$\mathbb{Z}_2 = 0$ characteristic of the A region and the $\mathbb{Z}_2 = 1$ region of the C region, the A-C boundary hosts additional edge conductivity as observed in the cAFM image (Fig. 2E). This only appears at specific low scanning voltages, (Fig. 2E and Supplementary Fig. 8 in Supplementary Information) and is experimentally challenging to detect. We excluded a geometric measurement artifact because the scans were repeated over various regions and in various directions. The additional boundary currents only appeared at low voltages and were electronic in origin as they responded to IV cycling. It is tempting to assign these as the quasi-1D edges at the A-C boundary predicted by topological theory. Indeed similar conductivity enhancement was observed at the edges of the terraces of Bi₂Se₃ where a physical vacuum ($\mathbb{Z}_2 = 0$) is present[45]. We add the disclaimer that, while the experiment is robust for the A-V boundaries, a great deal more experimental work is required to investigate the A-C boundaries to determine whether the "edge" 1D/2D conductivity is topological. This is because complex geometric, chemical, and crystallographic defects at the A-C boundary may potentially introduce trivial electronic states. Nevertheless, from a theoretical perspective, the existence of topological states at the A-C boundary is well-grounded, although we note these states may not be purely 1D, and may also contain a contribution from 2D TI states that wrap around the A-C boundary and appear as quasi-1D when viewed at the surface. Below we present theoretical modeling that supports the main experimental findings above for the A-V boundary, and also the secondary feature of 1D/2D states at A-C boundaries.

To develop a theory of the topological transition, and model the A-V and A-C interfaces, two approaches were used: density functional theory (DFT) and by solving a model Hamiltonian. Firstly, ab initio molecular dynamics (AIMD) simulations were conducted based on DFT to model the crystalline and amorphous components separately,

and the transition between states. DFT is a powerful tool, capable of performing the dual role of modeling glassy structures, and also topological Dirac states in crystals without empirical parameters. Although bulk chalcogenide glasses have been studied using similar methods[46,47], the past calculations did not include the spin–orbit interaction or surface slab models, so the question of the topological surface states remained unresolved. To directly compare the crystalline and non-crystalline surface/vacuum interface, the starting point was a large $Sb_2Te_3$ supercell (Fig. 3A). High-temperature AIMD was applied to generate a molten state at 1600 K which was quenched to 100 K into the "frozen" glass (for the details see the "Methods" section). The resulting structure of the glass compares very well with previously published pair-distribution functions and the model also correctly describes the finite-temperature vibrational density of states, implying a high level of realism in the model (see Supplementary Fig. 22). To model the 2D surface states at the A-V and A-C interfaces, an open boundary condition (OBC) was introduced in the $z$-direction by including a 15 Å vacuum region, and periodic boundary conditions (PBC) were applied in-plane. Using this calculation approach, the electronic density of states (DOS) for the glassy slab structure has a clear bandgap (Fig. 3A). In contrast, the crystalline starting cell shows in-gap states (Fig. 3B) attributed to the Dirac bands, as evident in the full band structure calculated (Fig. 3C). It is, therefore, valid, by induction, to conclude that the Dirac bands present in the crystalline state (Fig. 3C) must vanish after amorphization in order to result in the gap observed (Fig. 3A). This is expected if the system becomes a trivial insulator with $\mathbb{Z}_2 = 0$. This glassy-electronic transition is intrinsic (as it only required Sb–Te displacements and does not require impurities). However, similar calculations were performed including the Ga-impurity for completeness (Supplementary Fig. 26).

Additionally, by applying a minimal description of a TI using a model Hamiltonian, it is possible to show that, if a region becomes topologically trivial due to strong disorder, this will not only remove 2D states at the A–V boundary but also introduce new 1D/2D states at the A–C in ordered–disordered interfaces resembling some of the features observed experimentally (Fig. 2E). To describe the amorphous–crystalline boundary theoretically, a model Hamiltonian was introduced, starting from the accepted standard model of a fully crystalline TI[3,48]

$$H_{TI} = \begin{bmatrix} m(k) & Bk_z & 0 & Ak_- \\ Bk_z & -m(k) & Ak_- & 0 \\ 0 & Ak_+ & m(k) & -Bk_z \\ Ak_+ & 0 & -Bk_z & -m(k) \end{bmatrix} \quad (1)$$

where $m(k) = m_0 + m_1 k_z^2 + m_2(k_x^2 + k_y^2)$. The k.p. parameters are taken from ref. [49] and also reported in the Supplementary Information. To solve the eigenproblem, a multi-band envelope function and finite-element method were used. For a pristine, defect-free TI crystal surrounded by a physical vacuum, this model produces the well-known surface states as shown in Fig. 3D. Non-magnetic disorder was then introduced to this Hamiltonian to model the impact of the ion beam using a disorder potential:

$$V_{im}(r_i) = \sum_{i=1,N} v_i f(r - r_i) \quad (2)$$

where $v_i$ is the local potential which is assumed to be much larger than the bulk bandgap[27], and $f(r)$ is a range function (for simplicity assumed to be a cubic box potential, width of 3–5 Å). The total Hamiltonian was thus given by

$$H = H_{TI} + V_{im} \quad (3)$$

To model irradiation-induced damage similar to the experiment, we assume an ensemble of defects, with a probabilistic distribution in space according to a Gaussian in the vertical $z-$direction with a peak at 20 nm and width of 10 nm, approximating the experimental depth and Monte Carlo simulations for the Ga implantation. The resulting calculations show that the location of the surface state depends closely on the distribution of the defects, and the defect density $\rho_d$. For zero defects, the surface state only exists around the edges of the virtual crystal (Fig. 3D) at the physical crystal-vacuum boundary. A small percentage of defects does not modify this, and the surface state at the disordered region is still mostly intact and localized near the physical crystal-vacuum boundary (Fig. 3E). For high defect densities (Fig. 3F), however, the disordered region undergoes a quantum transition into an (Anderson) insulating layer. This converts the defective area into a "topological vacuum" such that the surface state now is shifted away from the physical vacuum and instead traces the boundaries of the new effective vacuum region. Notably, the surface state is not destroyed, instead, new states are formed underneath at the effective surface spanning the $\mathbb{Z}_2 = 1|0$ boundary. Importantly as these new states trace the outline of the disordered region, they will yield additional conductivity along the edge of an ion beam-patterned region. The new states at the A–C boundary include the standard 2D surface state but can also contain additional 1D states that form at the edge of the pattern, depending on the depth of the disordered region (Supplementary Fig. 28), thereby introducing a new source of conductivity not present in the unpatterned surface. This is qualitatively consistent with the cAFM measurements.

## Discussion

Both experiment and theory provide a strong argument that a $\mathbb{Z}_{2s} = 1 \rightarrow \mathbb{Z}_{2s} = 0$ transition occurs when crystalline $Sb_2Te_3$ is amorphized. The 2D topological surface states, which are present at the planar C–V boundary are destroyed by the non-periodic disorder which converts the region to an A–V boundary via ion beam processing. Thus our finding appears to be the first experimental proof that directly verifies earlier predictions concerning the level of topological protection against non-periodic non-magnetic disorder, namely that strong bulk disorder can lead to a quantum transition into a non-topological state[27]. Additionally, as a secondary point, there is some preliminary evidence for the existence of new "quasi-1D" states at the lateral A–C interfaces. Based on the theory, it is likely that these states contain a topological contribution, however, future work is needed to establish whether chemical and crystallographic effects introduce trivial electronic states. Even if these 1D A–C states are topological, they are a byproduct of the crystalline nature of one side of the interface, and thus do not modify our central claim about the non-topological nature of the 2D A–V interface. In terms of technological implications, ion beams were theoretically proposed as a promising method to engineer the topological conductivity[21], and our work shows how this is implemented in practice. Aside from lateral patterning, the ability to create vertical A–C boundaries with ion beam irradiation could potentially open a way to control buried topological states to yield functionality that is fundamentally different from junctions in traditional electronics. While electron irradiation has also been recently explored in TIs[50], there is a compelling reason to use ions rather than electrons for lateral and vertical patterning: the relative difference in inelastic scattering cross-sections makes it possible to create a much greater impact in a much smaller volume with ions to allow finer feature size and controlled amorphization. Feature size is a key goal for CMOS-compatible fabrication, where ion beams remain an "industry standard" tool[51,52]. Our initial work has mostly used a focused ion beam to implement patterning, as FIBs are readily available at most laboratories worldwide and enable rapid prototyping to test new principles in topological electronics. To this end, past work has established that FIBs can be used to achieve a variety of geometries and device functionalities in other topological materials[53,54], however, our work appears to be the first systematic study on the electronic effects of

ion-beam amorphization in topological insulators. However, in practice, it is not practical to use a FIB to pattern a large (wafer-sized) area. For large (cm²) scale irradiation, it is better to use standard accelerator-based broad ion beams in combination with lithography. In the Supplementary Information (Section 4.6), we have demonstrated that cm²-sized films can be amorphized using a 40 keV accelerator-based broad ion beam, showing that the underlying mechanism is identical to that in the FIB. This controlled functionality offers great opportunities for engineering the surface of TIs. In conventional silicon technology, the challenge is combining ion beams with photolithography to approach the 3–10 nm feature size. Our process was the first attempt and is far from optimized, but the cross-sectional STEM and Monte Carlo calculations show that we can already experimentally achieve vertical dimensions smaller than 20 nm, so with advanced lithography, this technique could foreseeably grant access to the sub-10 nm regime.

## Methods
The Supplementary Information file contains additional details on the methodology along with additional experimental results. Below a summary of the methodology is given.

### Crystal growth
The $Sb_2Te_3$ single-crystal growth method is described in ref. [40]. X-ray diffraction patterns confirmed the single-phase nature of $Sb_2Te_3$ possessing a rhombohedral crystal structure. The single crystal $Sb_2Te_3$ was cleaved using adhesive tape to expose a fresh and visually flat c-plane. Subsequent exfoliations for devices were done in the vacuum environment of the FEI HELIOS G3 microscope.

### Focused ion beam irradiation
The surface of $Sb_2Te_3$ was irradiated with a gallium focused ion-beam generated within an FEI HELIOS G3 CX microscope at an energy of 8–30 kV using an ion fluence ranging from $3.74 \times 10^{13}$ to $1.12 \times 10^{16}$ ions/cm². After the $Sb_2Te_3$ surface was irradiated with a Ga ion beam, electron backscattered diffraction images were obtained using a detector built into the microscope to confirm the amorphous nature of the $Sb_2Te_3$ surface. EBSD and general SEM imaging were performed on the same instrument. EBSD was conducted at 20 kV and 2.8 nA electron beam current.

### Conductive atomic force microscopy
The chip, affixed to which was the exfoliated sheet of $Sb_2Te_3$, was transferred in the air onto a magnetic atomic force microscopy holder for topography and local electrical characterizations using a commercial scanning probe microscopy (SPM) system (Cypher S, Asylum Research, US). Pt/Cr-coated conductive probes (ElectriMulti 75G, BudgetSensors, Bulgaria) with a radius of <25 nm were used for all the SPM measurements. The current mapping and local $I–V$ measurements were performed under conductive atomic force microscopy (cAFM) mode. The current mapping was acquired by scanning the probe across the selected areas with a DC bias between 2 and 2.5 V. During $I–V$ measurement, the probe was first engaged at a selected point on the film surface, followed by applying the sweeping bias function of 0 V to 1 V to -1 V to 0 V. Additional scans were performed up to 2.5 V.

### Device preparation and measurements
A FEI HELIOS G3 microscope was used for fabricating a four-probe resistance measurement device on a chip. A physical property measurement system (DynaCool, Quantum Design) was used for four-probe resistance and magnetoresistance measurements over a temperature range of 3–300 K.

### Computer simulations
Structural simulations and band-structure calculations were performed using density functional theory (DFT) in the Vienna Ab Initio Simulation Package (VASP) version 5.4.4[52]. The calculations used the generalized gradient approximation (GGA) to the exchange-correlation energy as implemented by Perdew et al. (GGA-PBE)[55]. The projector augmented-wave method was used[56–58] and the number of electrons treated as valence was 5 for Sb and 6 for Te. The cutoff energy for the plane-wave basis was 400 eV. The final structures were relaxed until forces converged to better than 0.02 eV/Å and the total energy convergence threshold was $10^{-6}$ eV. To describe the dispersion forces, the D3 Grimme method with zero damping was found to yield the best agreement with the experimental lattice parameters. For the crystalline models, we used a dense $k$-point grid equivalent to $x \times x \times n$ in the unit cell, where $x$ ranges from 1–16 $k$ points and $n = 16$ for 3D models and $n = 1$ for surface models. We found the Dirac surface states were already predicted using a single point in the crystalline models ($n = 1$, $x = 1$). Consequently, the amorphous model calculations were performed as single $\Gamma$-point calculations. For the surface models, a 15 Å vacuum slab was introduced to break the periodic boundary conditions in one direction, and slab layers were formed by cleaving at the van-der Waals gap of the bulk crystal structure which occurs at a specific (001) plane. For molecular dynamics, calculations were performed using ionic relaxation using the same convergence criteria outlined above, except with a single $K$-point at $\Gamma$. Large $Sb_2Te_3$ supercells ranging from $3 \times 3 \times 2$ to $4 \times 4 \times 1$ hexagonal cells were constructed using the crystalline starting model. These were heated to 1600 K in the simulation. The temperature was controlled by a Nose-isostat method to model the coupling to the heat bath. Melting occurred over the course of 3 ps. The molten structures were then quenched to 100 K over the course of 12 ps, using a time step of 2 fs. The mean-squared displacements and vibrational density of states of the DFT model were calculated by analyzing the trajectory using the nMoldyn software. This confirmed that the vibrational density of states accurately reproduced experimental measurements, and the ions did not diffuse at 100 K in the amorphous state. To generate fully optimized structures, additional 0 K relaxations with a higher convergence criterion were applied at the final step using a conjugate gradient method. The spin–orbit interaction was included when calculating the density of states and electronic band structure. Calculations were performed on the GADI supercomputer, which is part of the Australian National Computer Infrastructure.

## Data availability
All data are available in the main text or the Supplementary Information. The raw data is available upon request by contacting the corresponding author.

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

## Acknowledgements

We gratefully acknowledge Peter Evans and Michael Fuhrer for insightful discussions and comments at various stages of the project. We acknowledge Anton Le Brun for technical support with the reflectometry measurements. Abdulhakim Bake gratefully acknowledges a post-graduate research award (PGRA) from the Australian Institute for Nuclear Science and Engineering (AINSE). Q.Z. acknowledges the support of a Women in FLEET Fellowship. D.C. acknowledges the primary support of an ARC DECRA fellowship (DE180100314). This work was partially sup-ported by the ARC Centre for Excellence in Future Low Energy Electronic Technologies (CE170100039). J.K. and G.A. acknowledge funding from Australian Research Council Discovery Project (DP200102477). The work was made possible by the Electron Microscopy Centre at the University of Wollongong and used the FEI Helios G3 CX funded by the ARC LIEF grant (No. LE160100063), JEOL JEM-ARM200F funded by the ARC LIEF grant (No. LE120100104), and the JEOL JEM-F200 funded by the Uni-versity of Wollongong. We acknowledge the Australian Nuclear Science and Technology Organization for access to research facilities at the Centre for Accelerator Science and funding by the Australian Govern-ment through the NCRIS project. High-performance computing was undertaken using the Australian National Computing Infrastructure on the GADI supercomputer.

## Author contributions

A.B. performed the focused ion beam measurements, synthesized the devices, and co-wrote the manuscript. Q.Z. performed the conducting atomic force microscope measurements and N.V. contributed to the analysis of CAFM data. D.C. conceived the project, ran the DFT calcu-lations, and wrote the manuscript. X.W., W.Z., and Z.Y. prepared the single crystals. M.N. helped optimize all focused ion-beam and electron microscopy aspects. Z.P. performed implantation on the low-energy ion implanter at ANSTO. G.C. did the SRIM calculations. G.A. and A.N. contributed to the analysis of the transport data, and thin film growth. D.M. performed high-resolution STEM. C.H. and J.C. performed the model Hamiltonian calculations. R.L. and J.K. contributed to the inter-pretation of the results. All co-authors discussed the results and con-tributed to the manuscript.

## Competing interests

The authors declare no competing interests.
