## [Peer Review File · Nature Communications]

Top-Down Patterning of Topological Surface and Edge States using a Focused Ion BeamREVIEWER COMMENTS

Reviewer #1 (Remarks to the Author):

I read the paper "Top-Down Patterning of Topological Edge States using a Focused Ion Beam" with great interest. The proposed mechanism to write topological edge modes is interesting, and the FIB could be a good tool to do so. However, I would ask the authors to address a few issues before making a final recommendation:

- Why FIB? It is a wonderful tool, but it is not discussed in the manuscript why the FIB is important here. Likely, with (ebeam)-lithography and broad-beam milling one could achieve a similar goal – without the headache of Ga implantation (even though the authors have well shown this is not a problem). Especially with the application angle of the discussion, it would be important to mention that the FIB here prototypes but more scalable lithographic routes are clearly possible.

- Sharpen the main point. The paper tries to avoid the hard question, is there now a topological edge mode or not. Some sentences imply a very strong "yes" (This finding appears to be the first experimental proof which directly verifies earlier predictions concerning the level of topological protection against non-periodic disorder, namely that strong bulk disorder can lead to a quantum phase transition into a non-topological state.), while others are more cautious (...that may be premature, as complex chemical and crystallographic factors are certainly also present.). I think the more cautious approach is more defensible based on the data. This needs to be clarified by a clear sentence in the abstract, and this ambiguity needs to be removed from the manuscript.

- Phase change. It is not clear to me in what context this is a "phase change" technology. The ion-induced damage is a type of phase change of course, however it is one-way and highly unlikely to be perfectly reversed. Even then, it would require annealing at very high temperatures. Phase-change technology implies a directed and reversible transition from one phase to another as part of the operation. This word should either be removed, or well justified if I missed the point.

- Previous observations. This transition from trivial to non-trivial materials character due to FIB-amorphization was previously observed in Cd₃As₂ (<https://www.nature.com/articles/nature18276>). There, FIB was used to mill round patches similar to the squares which was shown to push the topological vacuum interface deeper into the physical bulk of the material.

- The transport should be explained. The data in Fig. 2F and its explanation is quite confusing. According to the authors, both sides of this bar are pristine. I wonder if that is true given that clearly FIB-deposited Pt/W was used to form the contacts. Due to the beam tails, likely some region in their vicinity, as well as underneath them, are amorphous too. More details on fabrication are important. Furthermore, in this simple parallel conductor picture the total conductivity should add, $S = S_{\text{bulk}} + 2S_{\text{surface}}$. It is strange that the conductivity is impacted so strongly, and even turns insulator-like by irradiating the top surface. The bottom surface cannot be irradiated, and hence I would expect $S = S_{\text{bulk}} + S_{\text{surface}}$ in the worst case. Adding an insulator/semi-conductor in parallel to a metal will not change its metallicity. This observation should be clearly explained.

- Clarify the cAFM measurements. There is an obvious anisotropy in the edge conduction (see Fig. 2 E). Two opposite surfaces are highly conductive, the other two are not as much. This is why the authors have rotated the image between Fig. 2E and the line profile in S8 (arguably that is misleading/cherry picking). What is the wave-like pattern within the amorphous phase? Is this an ion-beam induced surface rearrangement?

Reviewer #2 (Remarks to the Author):

In this experimental work, the authors use a focused ion beam to render regions of an Sb_2Te_3 crystal amorphous. Using theoretical and experimental probes, the authors then claim that amorphous Sb_2Te_3 is topologically trivial, which stands in contrast to pristine Sb_2Te_3 , which is a 3D Z₂ topological insulator (TI). Hence, boundaries between amorphous and ordered Sb_2Te_3 are expected to exhibit time-reversal-protected conducting surface states. The authors correspondingly observe enhanced conductance along boundaries between amorphous and ordered Sb_2Te_3 , consistent with this prediction.

I appreciate the thorough theoretical and experimental tests performed by the authors to support their findings, and find this work to be self-contained and compelling. One of the biggest outstanding issues with topological materials is their application to practical settings, and the nanoscale engineering of topological conducting channels performed in this work represents an important effort in that direction. Hence, I believe that this work rises to the level of impact necessary for publication in Nature Communications.

Before I can recommend this work for publication, I have a few comments and questions, which are enumerated below.

Major comments:

1. Throughout the manuscript, it is very unclear if the authors are claiming that the boundaries between the amorphous and ordered regions of Sb_2Te_3 are 1D or 2D. Frequently, the authors refer to "edges" and "channels," which convey a sense of 1D. However, if the mechanism underpinning the appearance of the conducting regions is topological in the sense claimed in this work, then the regions should in principle represent the 2D boundaries between topologically distinct 3D insulators. When resubmitting this manuscript, the authors should very carefully clarify the claimed dimensionality of all insulators, boundaries, and regions in precise terms (e.g. 1D, 2D, etc.) throughout both the main text and supplementary information, and state their widths if possible.

2. If the enhanced conductance signatures in the boundary regions arise from either 1D helical modes (which would require additional theoretical explanation) or 2D boundary Dirac cones, then the application of an external magnetic field should drastically lower the conductance of these regions by breaking time-reversal symmetry. The lowering of conductance under applied magnetism has historically represented an important validation of TI boundary physics, appearing in experiments ranging from [Konig, et al., Science (2007)] to [N. Shumiya, Md S. Hossain, ... M. Z. Hasan, Nat. Mater (2022)]. I recommend that the authors perform additional theoretical and experimental tests of the dependence of the boundary region conductance on external magnetic fields. This would provide the strongest indicator of whether the boundary states indeed arise from topologically protected 1D or 2D Dirac cones, as opposed to some other, non-topological mechanism.

Minor comments:

1. The authors refer to the website of Refs. 15 and 16 as the "Topological Quantum Database." The correct name of the database is the "Topological Materials Database."

2. Close to where the database in Ref. 15 is discussed, the authors refer to Sb_2Te_3 as "a rare example of a strong TI," citing Ref. 1. However, as demonstrated in Refs. 15 and 16, strong 3D TIs are not nearly as rare among known materials as originally thought (though members of the Bi_2Se_3 family like Sb_2Te_3 still represent some of the best 3D TI candidates for experimental applications).

3. Early in the manuscript, the authors employ a footnote with regard to the strong 3D Z2 index. I am not sure if footnotes are permitted in the Nat. Comm. style, and in either case, it would be better for accessibility to perhaps incorporate this comment in some form into the body of the main text.

4. In the discussion of Fig. 3c in the main text, the authors state that a surface band gap could be induced by shifting topological Dirac bands away from the Fermi level. However, topological surface states do not just consist of isolated Dirac cones, but also have spectral flow, such that the Dirac cones near the Fermi level connect to other Dirac cones at higher and lower energies. Spectral flow in fact prevents a shift of the surface chemical potential from removing some form of Dirac band on the surface of a 3D TI [see Ref. 1]. I recommend that the authors remove the "shifting" part of this argument in the revised manuscript, or clarify it, to more properly accommodate the theoretical topological notion of spectral flow that accompanies surface Dirac band in 3D TIs.

5. In section 5 of the supplementary information, and its summary in the main text, the authors should clarify that they are modeling the effects of nonmagnetic disorder.

Reviewer #3 (Remarks to the Author):

The authors in "Top-Down Patterning of Topological Edge States using a Focused Ion Beam" developed a new method to spatially pattern amorphous regimes in Sb₂Te₃, a 3D topological insulator, aiming to pattern the topological surface conduction channels into topological circuitry. This work contains adept details about Ga transport and materials characterization of Sb₂Te₃. This method could bring a simple solution (or a step closer) to pattern topological channels into real circuitry. I would recommend its publication if my following questions/concerns are addressed. The main weakness in this work, in my opinion, is the lack of sufficient experimental data to support the annihilation of SS by introducing sufficient disorder with FIB. While sufficient disorder in TIs is shown in the literature to be able to annihilate the topological invariant, the FIB method introduces amorphous disorders only at the top <20nm and the surface states (SS) might very well still be preserved underneath the amorphous regime. In order to pattern/control the topological channels on the selected areas, it is desired to annihilate the SS and leave only the edge states. In a sense, the SS might be pushed deeper into the bulk by the scrambled surface crystal lattices. In fact, the DOS graph in Figure 3F shows the SS is wrapping around the amorphous regime and contradicting to author's goal/claim to be able to pattern topological channels (It might work on sufficiently thin samples (~10nm) but it should be demonstrated). The range of disorder strength in the function of depth and its bulk conductivity are also not well understood/shown. I would recommend calculating the SS distribution as a function of depth and/or bandstructure for crystalline/amorphous slabs. Sufficient experiment evidence for the crystalline regime underneath the amorphous area should be presented. The conductive AFM data only show the conductivity on the surface and is not sufficient to probe the SS underneath. The main text on pg. 3. mentioned "Shubnikov-de Haas oscillations, a non-trivial Berry phase,..." but these data are not presented in main or supplementary for pristine and after FIB patterned. It would be helpful to characterize and compare the SS electron transport before and after the FIB pattern if the Fermi level of the material is sufficiently close or within the bulk gap. Regarding the range and stopping power of Ga ion used in the experiment. Could you show the Ga ion Bragg peak for the energy used in the experiment? The distribution of Ga ions shown in Figure S3 is very noisy due to the low sensitivity of EDS (TOF-SIM experiment as a function of depth might help clarify the distribution of the Ga ions). How well is the morphology as a function of depth? Since the stopping of Ga might be concentrated in a certain depth, the morphology might be significantly different at the Bragg peak vs at the surface. Transport data showing significantly high resistivity at the patterned regime would be convincing. This method relies on large resistivity

different and sharp spatial interface between amorphous and crystalline states. How robust are these amorphous states against temperature annealing? Transport data as a function of FIB ion energy, fluences, and temperature annealing would be very helpful.

Review comments are in black. Our response is in blue.

Reviewer #1 (Remarks to the Author):

I read the paper “Top-Down Patterning of Topological Edge States using a Focused Ion Beam” with great interest. The proposed mechanism to write topological edge modes is interesting, and the FIB could be a good tool to do so. However, I would ask the authors to address a few issues before making a final recommendation:

1.1. Why FIB? It is a wonderful tool, but it is not discussed in the manuscript why the FIB is important here. Likely, with (ebeam)-lithography and broad-beam milling one could achieve a similar goal – without the headache of Ga implantation (even though the authors have well shown this is not a problem). Especially with the application angle of the discussion, it would be important to mention that the FIB here prototypes but more scalable lithographic routes are clearly possible.

We agree with the reviewer that we can clarify this better. In response, we have added the following to the conclusion section.

“In terms of technological implications, ion beams were theoretically proposed as a promising method to engineer the topological conductivity²¹, and our work shows how this is implemented in practice. While electron irradiation has also been recently explored in TIs, there is a compelling reason to use ions rather than electrons is the difference in inelastic scattering cross-sections. It is possible to create a much greater impact in a much smaller volume with ions to allow finer feature size and controlled amorphization. Feature size is a key factor for CMOS-compatible fabrication, where ion beams remain an “industry standard” tool. Our initial work has mostly used a focused ion beam to implement patterning, as FIBs are readily available at most laboratories worldwide and enable rapid prototyping to test new principles in topological electronics. To this end, past work has established that FIBs can be used to achieve a variety of geometries and device functionalities in topological materials^{48,49}, however our work appears to be the first systematic study on the electronic effects of ion-beam amorphization in topological insulators. However, in practice, it is not practical to use a FIB to pattern a large (wafer-sized) area. For large (cm²) scale irradiation, it is better to use standard accelerator-based broad ion beams in combination with lithography. In the supplemental (Section 4.6), we have demonstrated that cm²-sized films can be amorphized using a 30 keV accelerator-based broad ion beam, showing that the underlying mechanism is identical to that in the FIB. This controlled functionality offers great opportunities for engineering the surface of TIs.”

Reflecting that the focused ion beam is only one method used in the work, we have rephrased the title to clarify that the important factor is “ion beam” not necessarily “focused ion beam”.

1.2. Sharpen the main point. The paper tries to avoid the hard question, is there now a topological edge mode or not. Some sentences imply a very strong “yes” (This finding appears to be the first experimental proof which directly verifies earlier predictions concerning the level of topological protection against non-periodic disorder, namely that strong bulk disorder can lead to a quantum phase transition into a non-topological state.), while others are more cautious (...that may be premature, as complex chemical and crystallographic factors are certainly also present.). I think the more cautious approach is more defensible based on the data. This needs to be clarified by a clear sentence in the abstract, and this ambiguity needs to be removed from the manuscript.

We have revised the manuscript to stress that the main point of our paper is the removal of the 2D topological surface state by the introduction of non-periodic non-magnetic disorder.

In the comment above it appears that the reviewer is conflating two separate claims: one concerning the 2D surface state at the amorphous-vacuum boundary (where the evidence is very clear) and one concerning the 1D state at the amorphous-crystal boundary (where the evidence is more preliminary). As noted by the second reviewer (comment 2.2), we inadvertently caused confusion by using the term “edge” mode to refer to both 1D and 2D topological states, both at the amorphous-crystalline and amorphous-vacuum boundary. To communicate this better, we have now defined a more precise nomenclature early in the paper, which also clearly points out the different types of interfaces present in the experiment: “We find that, with ion beam processing, we can study and control three distinct types of interfaces in a topological insulator, each with distinctive electronic properties. The nomenclature used here to describe these three interfaces are: A-V for amorphous-vacuum, C-V for crystalline-vacuum and A-C for amorphous-crystal boundaries. Furthermore, we observe modified conductivity from 2D topological “surface states” and 1D “edge” regions”.

We have added a section emphasizing that the claims around the A-V boundary are the most robust: “The experimental evidence for the insulating 2D A-V boundaries is extremely robust, and it is also supported by the density functional theory (described in a later section) which explains this on the basis of a topological transition to $\mathbb{Z}_2=0$. The observation of the highly insulating A-V 2D region is the main claim of our work as it appears consistently in all patterned regions above the critical fluence level. There are, however, also interesting secondary effects that occur at many of the 1D A-C boundaries which warrant discussion... We add the disclaimer that, while the experiment is robust for the A-V boundaries, a great deal more experimental work is required to investigate the A-C boundaries to determine whether the “edge” 1D/2D conductivity is topological. This is because complex geometric, chemical and crystallographic defects at the A-C boundary may potentially introduce trivial electronic states. Nevertheless, from a theoretical perspective, the existence of topological states at the A-C boundary is well-grounded, although we note these states may not be purely 1D, and may also contain a contribution from 2D TI states that wrap around the A-C boundary and appear as quasi-1D when viewed at the surface “

In the final conclusion we have re-stated our main claim explicitly: “The 2D topological surface states, which are present at the planar C-V boundary are destroyed by the non-periodic disorder which converts the region to an A-V boundary via ion beam processing. Thus our finding appears to be the first experimental proof that directly verifies earlier predictions concerning the level of topological protection against non-periodic non-magnetic disorder, namely that strong bulk disorder can lead to a quantum transition into a non-topological state. Additionally, as a secondary point, there is some preliminary evidence for the existence of new “quasi-1D” states at the lateral A-C interfaces. Based on the theory, it is likely that these states contain a topological contribution, however future work is needed to establish whether chemical and crystallographic effects introduce trivial electronic states. Even if these A-C 1D states are topological, they are a byproduct of the crystalline nature of one side of the interface, thus do not modify our central claim about the nature of the 2D A-V interface “

1.3. Phase change. It is not clear to me in what context this is a “phase change” technology. The ion-induced damage is a type of phase change of course, however it is one-way and highly unlikely to be perfectly reversed. Even then, it would require annealing at very high temperatures. Phase-change technology implies a directed and reversible transition from one phase to another as part of the operation. This word should either be removed, or well justified if I missed the point.

We have followed the reviewers’ suggestion and removed the word from the abstract. Also see Reviewer comment 3.4.

1.4. Previous observations. This transition from trivial to non-trivial materials character due to FIB-amorphization was previously observed in Cd₃As₂ (<https://www.nature.com/articles/nature18276>). There, FIB was used to mill round

patches similar to the squares which was shown to push the topological vacuum interface deeper into the physical bulk of the material.

We thank the reviewer for bringing this to our attention and have cited this paper. There are, however, some significant differences from our work in that the previous study was conducted on a topological Dirac semimetal, not a topological insulator, and as such the nature of the topological surface/edge states is rather different. Furthermore, in that study, the damage layer was a secondary and unintentional side effect of the FIB, whereas in our work we have deliberately and systematically sought to control the disorder region, both laterally and vertically, to enable top-down patterning.

1.5. The transport should be explained. The data in Fig. 2F and its explanation is quite confusing. According to the authors, both sides of this bar are pristine. I wonder if that is true given that clearly FIB-deposited Pt/W was used to form the contacts. Due to the beam tails, likely some region in their vicinity, as well as underneath them, are amorphous too. More details on fabrication are important. Furthermore, in this simple parallel conductor picture the total conductivity should add, $S=S_{\text{bulk}}+2S_{\text{surface}}$. It is strange that the conductivity is impacted so strongly, and even turns insulator-like by irradiating the top surface. The bottom surface cannot be irradiated, and hence I would expect $S=S_{\text{bulk}}+S_{\text{surface}}$ in the worst case. Adding an insulator/semi-conductor in parallel to a metal will not change its metallicity. This observation should be clearly explained.

We have added more details on the device fabrication in the supplemental. The way the sample is cut avoids amorphising the main surface and the detectable area of the surface remains crystalline, as shown by the EBSD maps (see Supplemental Figures 14, 15).

Regarding the question about parallel conduction, it is important to note that the Pt electrodes only contact the surface, and not the bulk crystalline section of the device. In this case, any current flowing through the bulk must first pass through the surface barrier which introduces an additional series resistance, as per the lumped-element model described below, accounting for some finite leakage (R_L) in the amorphous region. The total resistance is

$$R_T = \frac{R_S (2R_L + R_B)}{R_S + R_B + 2R_L}$$

Using realistic values for the surface, leakage and bulk resistivities, the model reproduces the observations in the FIB device measurements with a sufficient degree of accuracy. Also note that in addition to the original experiments on microscale Sb_2Te_3 FIB devices, we have now added complementary measurements on 10-30 nm thick Sb_2Te_3 films grown on an insulating substrate (sapphire). These provide a better measurement of the amorphous resistance in that the parallel bulk channel is completely suppressed, and there is no leakage resistance. That data shows that the resistance of the amorphous phase increases into the megaohm range, and using this number to constrain the terms in the lumped-element model for the FIB devices, a self-consistent picture emerges as described in Supplemental Section 4.5.

Figure R1. Simplified lumped-element model of an ion-beam irradiated device.

1.6. Clarify the cAFM measurements. There is an obvious anisotropy in the edge conduction (see Fig. 2 E). Two opposite surfaces are highly conductive; the other two are not as much. This is why the authors have rotated the image between Fig.2E and the line profile in S8 (arguably that is misleading/cherry picking). What is the wave-like pattern within the amorphous phase? Is this an ion-beam induced surface rearrangement?

It is important to note that, while there is some anisotropy in the “1D edge states” at the C-A boundary, there is no detectable anisotropy for the effect that occurs at the “2D” A-V boundary (where the latter is the most robust finding in our work, and significant in itself, as we have aimed to emphasize in the revision).

Concerning the anisotropic effect on the pattern edge conductivity for the “1D” C-A boundaries: This is due to the nature of conductive atomic force microscopy (CAFM), of which the conductive tip was scanned along the X-axis of Figure 2E (leftwards and rightwards). At the edges of the pattern where there are stages/steps, the contact area of the tip with sample could become unstable (with a higher or lower contact area depending on the surface condition). This causes the observed anisotropy in Figure 2E.

To exclude the factors from scanning probe microscopy, we also performed additional experiments as detailed below:

(1) Change the scanning direction.

As shown in **Figure R2**, the left (**Figure R2a**) and right (**Figure R2b**) CAFM images are collected from the same pattern, but the scanning angle has been rotated by 90 degrees (as indicated by the tip position and fast/slow scanning directions). The numbers inside the figures label the edges before and after the rotation. It is found that, for either scan, the edges of the pattern show higher conductivity and the etched pattern areas show relatively lower conductivity. In Figure R2a, edges 1 and 3 show relatively higher current than edges 2 and 4; however, in Figure R2b, edges 1 and 4 both show higher conductivity. Please note the scale of the current value is quite small (within 15 pA) and the probe-sample contact issue (as mentioned above) during scanning could cause such sensitive variation. Also, note that the central 2D region of the square has no measurable current in either configuration. This is the main claim of our work.

Figure R2. Current mapping using 1V DC bias for an individual irradiated box. (a) CAFM scans with fast scanning direction parallel with the probe; (b) CAFM scans with fast scanning direction perpendicular to the probe.

(2) Collect CAFM images with edges at different angles

Figure R3 shows a triangular shaped logo, which includes several edge lines at different angles. It can be observed that all the edges show relatively higher conductivity (including the small circles inside the pattern), while the etched pattern shows lower conductivity.

Figure R3. (a) Topography and (b) current mapping of an individual irradiated logo pattern.

(3) Collect I-V at the edge

As mentioned above, the steps could cause artifacts due to the change of tip/sample surface contact area during scanning. Thus, to confirm the observed high conductivity from CAFM is real, we have also performed I-V measurements at the irradiated (amorphous) areas, A-C edges, as well as the unirradiated (crystalline) areas, by loading the probe on the selected points prior to applying bias. The results in Figure R4 confirm that the high conductivity from the edges is real, and the irradiated (amorphous) area shows much lower current than both the A-C edges and crystalline regions.

Figure R4. I-V measurements on the several (a) irradiated (amorphous) areas, (b) A-C edges, and (c) unirradiated (crystalline) areas.

Regarding “cherry picking of data”, we firmly believe in presenting all data. In the submitted manuscript, we intended to give a strong and clear message to the readers by presenting the most typical and clear observations (i.e., Figure 2E, Figure S9, and Figure S10). That is, the middle of the irradiated patterns is always insulating, and the edges of the irradiated patterns typically show higher conductivity, for some bias voltages. This conclusion is based on tens of repeated CAFM scans on various patterns (as shown above). Moreover, the rotation of Figure S10 in the supplemental is just to show a horizontal current line scan that matches with the horizontal x-axis of the current profile figure. We also understand the referee’s concern regarding reproducibility and have added some of the discussion above to the supporting information.

Reviewer #2 (Remarks to the Author):

In this experimental work, the authors use a focused ion beam to render regions an Sb_2Te_3 crystal amorphous. Using theoretical and experimental probes, the authors then claim that amorphous Sb_2Te_3 is topologically trivial, which stands in contrast to pristine Sb_2Te_3 , which is a 3D Z2 topological insulator (TI). Hence, boundaries between amorphous and ordered Sb_2Te_3 are expected to exhibit time-reversal-protected conducting surface states. The authors correspondingly observe enhanced conductance along boundaries between amorphous and ordered Sb_2Te_3 , consistent with this prediction.

I appreciate the thorough theoretical and experimental tests performed by the authors to support their findings, and find this work to be self-contained and compelling. One of the biggest outstanding issues with topological materials is their application to practical settings, and the nanoscale engineering of topological conducting channels performed in this work represents an important effort in that direction. Hence, I believe that this work rises to the level of impact necessary for publication in Nature Communications.

Before I can recommend this work for publication, I have a few comments and questions, which are enumerated below.

Major comments:

2. 1. Throughout the manuscript, it is very unclear if the authors are claiming that the boundaries between the amorphous and ordered regions of Sb₂Te₃ are 1D or 2D. Frequently, the authors refer to “edges” and “channels,” which convey a sense of 1D. However, if the mechanism underpinning the appearance of the conducting regions is topological in the sense claimed in this work, then the regions should in principle represent the 2D boundaries between topologically distinct 3D insulators. When resubmitting this manuscript, the authors should very carefully clarify the claimed dimensionality of all insulators, boundaries, and regions in precise terms (e.g. 1D, 2D, etc.) throughout both the main text and supplementary information, and state their widths if possible.

This is a useful insight. We agree that more precise nomenclature would help clarify the new insights provided by our work. We now have defined the nomenclature in the introduction (see Reviewer comment 1.2). The situation at the C-A boundary is complicated, and, while we agree that the 2D boundary between distinct insulators is generically present, in our theory section we also show that 1D edge states can be present in special cases. Direct measurements of the widths is challenging, however in the theory sections we have given estimates for the decay constants of the different types of edge/surface states.

2.2. If the enhanced conductance signatures in the boundary regions arise from either 1D helical modes (which would require additional theoretical explanation) or 2D boundary Dirac cones, then the application of an external magnetic field should drastically lower the conductance of these regions by breaking time-reversal symmetry. The lowering of conductance under applied magnetism has historically represented an important validation of TI boundary physics, appearing in experiments ranging from [Konig, et al., Science (2007)] to [N. Shumiya, Md S. Hossain, ... M. Z. Hasan, Nat. Mater (2022)]. I recommend that the authors perform additional theoretical and experimental tests of the dependence of the boundary region conductance on external magnetic fields. This would provide the strongest indicator of whether the boundary states indeed arise from topologically protected 1D or 2D Dirac cones, as opposed to some other, non-topological mechanism.

We agree that magnetic fields are a useful tool to differentiate between the 1D helical and 2D states at the A-C boundary, both of which potentially appear at the A-C boundary. We have added magnetotransport experiments in the supplemental (see comment 3.4), however, those experiments do not only probe the A-C boundary, but also the conductivity of the A and C region, making it difficult to draw a firm conclusion on the reviewers point. Substantial re-engineering of the transport experiments would be required to isolate the A-C contribution, and this is well beyond the scope of the current work, and will be the subject of future work. We agree, in principle, that magnetic fields are the ideal tool to differentiate the 1D and 2D states. We have performed preliminary theory, showing how the two different scenarios respond to fields, by using the following Hamiltonian:

$$H = H_{TI} + H_Z \otimes \mathbb{I}_2$$
$$H_Z = g_{\parallel} \mu_B \sigma_z B_z + g_{\perp} \mu_B (\sigma_x B_x + \sigma_y B_y)$$

The results depend on the field direction and magnitude as shown in the Figure R5 below. Generally, the 2D states survive; however, the 1D states can be removed by applying the field in specific directions. However, it has not yet been possible to verify this experimentally as it would require a substantial redesign of the devices and apparatus, and this will be the target of future work.

Figure R5. Surface states distribution in the presence of a magnetic field. (a) A magnetic field applied along the y-direction. (b) A magnetic field applied along the x-direction. (c) a magnetic field applied along the z-direction.

Minor comments:

2.1. The authors refer to the website of Refs. 15 and 16 as the “Topological Quantum Database.” The correct name of the database is the “Topological Materials Database.”

Thank you, this has been corrected.

2.2. Close to where the database in Ref. 15 is discussed, the authors refer to Sb₂Te₃ as “a rare example of a strong TI,” citing Ref. 1. However, as demonstrated in Refs. 15 and 16, strong 3D TIs are not nearly as rare among known materials as originally thought (though members of the Bi₂Se₃ family like Sb₂Te₃ still represent some of the best 3D TI candidates for experimental applications).

We have replaced “rare example” with “good example.”

2.3. Early in the manuscript, the authors employ a footnote with regard to the strong 3D Z₂ index. I am not sure if footnotes are permitted in the Nat. Comm. style, and in either case, it would be better for accessibility to perhaps incorporate this comment in some form into the body of the main text.

We have revised the formatting standards to be compliant with the Nat. Comm. style and removed the footnote.

2.4. In the discussion of Fig. 3c in the main text, the authors state that a surface band gap could be induced by shifting topological Dirac bands away from the Fermi level. However, topological surface states do not just consist of isolated Dirac cones, but also have spectral flow, such that the Dirac cones near the Fermi level connect to other Dirac cones at higher and lower energies. Spectral flow in fact prevents a shift of the surface chemical potential from removing some form of Dirac band on the surface of a 3D TI [see Ref. 1]. I recommend that the authors remove the “shifting” part of this argument in the revised manuscript, or clarify it, to more properly accommodate the theoretical topological notion of spectral flow that accompanies surface Dirac band in 3D TIs.

We thank the reviewer for this point and agree. We have removed the discussion of “shifting”, partly because a full discussion of the spectral flow would be difficult within the length constraints.

2.5. In section 5 of the supplementary information, and its summary in the main text, the authors should clarify that they are modeling the effects of nonmagnetic disorder.

We have added “non-magnetic” in Section 5 and the summary in the main text.

Reviewer #3

The authors in “Top-Down Patterning of Topological Edge States using a Focused Ion Beam” developed a new method to spatially pattern amorphous regimes in Sb₂Te₃, an 3D topological insulator, aiming to pattern the topological surface conduction channels into topological circuitry. This work contains adept details about Ga transport and materials characterization of Sb₂Te₃. This method could bring a simple solution (or a step closer) to pattern topological channels into real circuitry. I would recommend its publication if my following questions/concerns are addressed.

3.1. The main weakness in this work, in my opinion, is the lack of sufficient experimental data to support the annihilation of SS by introducing sufficient disorder with FIB. While sufficient disorder in TIs is shown in the literature to be able to annihilate the topological invariant, the FIB method introduces amorphous disorders only at the top <20nm and the surface states (SS) might very well still be preserved underneath the amorphous regime.

(It might work on sufficiently thin samples (~10nm) but it should be demonstrated).

We have now added data to the supplemental for 10-30 nm thick MBE-grown films (Sb₂Te₃) exposed to an ion beam showing that this does indeed work. If the film is entirely amorphized the resistance rises from a few ohms into the megaOhm range, and the entire layer is disordered with no residual crystallinity (see TEM, XRR, XRD in the Supplemental).

3.2. In order to pattern/control the topological channels on the selected areas, it is desired to annihilate the SS and leave only the edges states. In a sense, the SS might be pushed deeper into the bulk by the scrambled surface crystal lattices.

We agree with the reviewer that the 2D surface state for an amorphous-crystalline slab (such as those presented in our initial work) can potentially be “pushed” deeper into the sample. However, in practice this means that it will respond very differently to an electrode or nearby gate voltage, which still constitutes a form of engineering. Furthermore, the only alternative way to reposition the state in real-space would be physically to cut or cleave the sample, which is clearly not as convenient as the ion-beam method we describe. Thirdly, as per reviewer comments 2.2 and 2.3, there are also theoretical scenarios where new 1D edge states are introduced which are not present in the naked surface, and constitute an additional avenue for engineering.

3.3. In fact, the DOS graph in Figure 3F shows the SS is wrapping around the amorphous regime and contradicting to author’s goal/claim to be able to pattern topological channels

We draw the reviewer’s attention to our additional calculations (Supplemental 6.2) which shown that, along with the 2D state, there is also a new 1D state that emerges in some scenarios, and the overall conductivity, and gateability of the conduction pathway will be modified in the patterned region. We also draw the reviewer’s attention to past theory surrounding the potential engineering advantages of repositioning the surface state, see: Sacksteder, V., Ohtsuki, T. & Kobayashi, K. Modification and control of topological insulator surface states using surface disorder. *Physical Review Applied* 3, 064006 (2015). In fact, the only other viable way to reposition the surface state would be through physically cutting or cleaving the crystal, so we believe that our method is a valid and promising form of engineering, as it allows a single surface to be patterned into many different regions.

3.4. The range of disorder strength in the function of depth and its bulk conductivity are also not well understood/shown. I would recommend calculating the SS distribution as a function of depth and/or band structure for crystalline/amorphous slabs

Additional calculations have been performed as in the Figure R6 using the model Hamiltonian method and have been added to the Supplemental Section 6.2 and mentioned in the text.

Figure R6. Deformation of surface states with increased disorder depth.

3.5. Sufficient experiment evidence for the crystalline regime underneath the amorphous area should be presented.

The high-resolution STEM images clearly show the crystalline lattice fringes. Furthermore, in the revised Supplemental we have also added XRD for an irradiated crystal, as used for the FIB devices, and also for the MBE-grown thin films, showing the existence of Bragg reflections, indicating the crystalline underlayer.

3.6. The conductive AFM data only show the conductivity on the surface and is not sufficient to probe the SS underneath.

We thank the referee for this comment, however we refer to the first reviewer's comment (1.5) showing that, because the surface has a finite leakage resistance, and there is a bulk conductive channel in the crystals, there will be some sensitivity to bulk / buried conductivity channels.

We also would like to clarify that the CAFM measurement actually collects the current data in the crystal / film thickness direction, rather than the surface, as explained in Figure R7. During a CAFM measurement, the sample is mounted on a conductive holder where the bias is applied through the substrate (or sample) towards the probe. Thus, the conductive data actually contains information about the transport through the entire sample thickness (i.e., surface and underneath). Thus, as illustrated in CAFM working principle Figure R7 below, the CAFM measurement will indeed measure current from the buried regions underneath the surface, provided the leakage resistance through the amorphous region is not infinite.

Figure R7. Conductive atomic force microscopy (CAFM) working principle

3.7. The main text on page 3 mentioned “Shubnikov-de Haas oscillations, a non-trivial Berry phase,...” but these data are not presented in main or supplementary for pristine and after FIB patterned. It would be helpful to characterize and compare the SS electron transport before and after the FIB pattern if the Fermi level of the material is sufficiently close or within the bulk gap.

We have now added the magnetoresistance (ρ_{xx}) measurements for the pristine/ion-beam exposed samples. This shows clear SdH oscillations in the pristine device similar to those found in past bulk Sb_2Te_3 with p-type conductivity. The ion-beam exposed device has a much higher resistance, with vanishingly small MR and no detectable oscillations. In the final (non-anonymous) version of the manuscript we will also add the reference to the past published measurements of the bulk crystals, which are very similar to the pristine FIB devices.

Figure R8. Transport measurements of the FIB fabricated device.

3.8. Regarding the range and stopping power of Ga ion used in the experiment. Could you show the Ga ion Bragg peak for the energy used in the experiment? The distribution of Ga ions shown in Figure S3 is very noisy due to the low sensitivity of EDS (TOF-SIM experiment as a function of depth might help clarify the distribution of the Ga ions). How

well is the morphology as a function of depth? Since the stopping of Ga might be concentrated in a certain depth, the morphology might be significantly different at the bragg peak vs. at the surface.

The Bragg peak is presented in the Monte Carlo calculations in the supplemental. To minimize the effects of Ga ion doping, and associated complexities, we have deliberately used very light Ga doses resulting in atomic percents < 1 % which makes this challenging to detect experimentally with EDS/RBS. However, we note that the depth of the amorphous region is an equally valid measure of the ion penetration depth, since the damage that is apparent in the STEM images is a direct measure of the penetration depth and matches the detailed Monte Carlo calculations within ± 2 nm. The STEM images also show that the morphology of the A-C interface is indeed complex.

3.9. Transport data showing significantly high resistivity at the patterned regime would be convincing. This method relies on large resistivity difference and sharp spatial interface between amorphous and crystalline states.

It is slightly unclear as to what the reviewer means here. Transport data is already presented in Fig 2, noting the difference between the irradiated/unirradiated samples, and complementary CAFM measurements have been performed for a range of ion fluences showing large changes in the patterned regions. We have also added transport data for the MBE-grown films showing a transition from the Ohm scale to megaOhm scale for ion-beam irradiated films.

3.10. How robust are these amorphous states against temperature annealing?

We have added in-situ annealing data in the supplemental as shown in Figure R9 below. A moderate heat treatment to 400 °C causes recrystallization.

Figure R9. In-situ characterization of a Sb₂Te₃ bar (a) before irradiation, (b) post-irradiation and (c) post-heating.

3.11. Transport data as a function of FIB ion energy, fluences, and temperature annealing would be very helpful.

We point out that the Figure 2 shows a series of fluences, where the conductivity is measured. One of the challenges of trying to do this across multiple Sb_2Te_3 crystals is the variability in the carrier concentration/transport parameters between crystals even those from the same growth ingot. Thus, it is highly beneficial to study a series of fluences on a single region as shown in Fig 2.

REVIEWERS' COMMENTS

Reviewer #1 (Remarks to the Author):

This is my second reading of the manuscript "Top-Down Patterning of Topological Surface and Edge States using an Ion Beam". It touches the highly topical subject of electronics applications of topological states, and I believe in its current version it can be published at Nature Communications. The authors have responded to my and the other reviewers questions sufficiently, in my view.

Just small points to consider:

- 1) I would put the "Focused" back into the title. Its true that scalability and physics requires an ion beam of any kind. But the presented data were all taken using a FIB and are not necessarily transferrable to broad beams (e.g. heating and non-linear interactions). Lastly, the research community works mostly with FIBs so it would attract readers also on the technical side.
- 2) It may be wise to not discredit the potential/expected topological state at the A/C interface, now buried in the bulk (as mentioned by all reviewers). You are proposing a new guiding principle to fabricate topological devices, and the topological protection is what sets this fundamentally apart from traditional electronics/channels in 2DEGs. This is a feature not a bug! I would advise you not to treat it as an issue to be argued away, but rather as a unique and novel aspect of disorder-designing topological electronics.

Reviewer #2 (Remarks to the Author):

In the revised manuscript, the authors have performed a thorough and commendable amount of additional experiments and theoretical calculations to further support their claims and to respond to the reviewers' comments. The authors have responded to all of my previous comments except the request to perform additional experiments with external magnetism to test the Dirac nature of the AC boundary states and their protection by time-reversal symmetry. However, the authors have persuasively stated that these experiments would be complex and beyond the scope of the present work, and have also performed numerical calculations simulating the effects of the external field. I am therefore satisfied with the authors' responses to myself and the other referees, provided that the external magnetic field calculations are included in the final version of the manuscript in either the main text or the Supplementary Material (SM).

Because I previously found this work to rise to the level of impact necessary for publication in Nature Communications, and because I find the revised manuscript to now be clear and compelling – especially with regard to the differences between interfaces and their dimensionality – I recommend this work for publication after the authors address a few final, minor points, which I have enumerated below.

1. In section 4.2 of the SM, the word mapping is misspelled as "mampping."
2. In the same section of the SM, the word rectangular is misspelled later in the same paragraph.
3. In section 6.3 of the SM, the authors briefly refer to the location of the 1D state as a "hinge." To avoid the ongoing debate as to the nature and presence of 1D "hinge" states in 3D topological insulators. I recommend that the authors change this and all other instances of the word "hinge" to "edge" throughout the main text and SM

Reviewer #3 (Remarks to the Author):

My concerns are fully addressed. Thank you for the additional data and calculations, they are very helpful. In addition, the new calculation results, showing emerging 1D edge channels as surface states are pushed deeper into the bulk, are particularly interesting.

Review comments are in black. Our response is in blue.

Reviewer #1 (Remarks to the Author):

This is my second reading of the manuscript "Top-Down Patterning of Topological Surface and Edge States using an Ion Beam". It touches the highly topical subject of electronics applications of topological states, and I believe in its current version it can be published at Nature Communications. The authors have responded to my and the other reviewers questions sufficiently, in my view.

We thank the reviewer for their positive comments and are pleased that they agree the work is suitable for publication.

Just small points to consider:

1) I would put the "Focused" back into the title. Its true that scalability and physics requires an ion beam of any kind. But the presented data were all taken using a FIB and are not necessarily transferrable to broad beams (e.g. heating and non-linear interactions). Lastly, the research community works mostly with FIBs so it would attract readers also on the technical side.

We are grateful for the suggestion. Note that the supplemental material does contain some data taken with "unfocused" ion beams, however the main results in the manuscript are indeed obtained with a FIB. We have added "focused" back into the title.

2) It may be wise to not discredit the potential/expected topological state at the A/C interface, now buried in the bulk (as mentioned by all reviewers). You are proposing a new guiding principle to fabricate topological devices, and the topological protection is what sets this fundamentally apart from traditional electronics/channels in 2DEGs. This is a feature not a bug! I would advise you not to treat it as an issue to be argued away, but rather as a unique and novel aspect of disorder-designing topological electronics.

We agree with the reviewer. We have added one sentence expressing this point in the discussion: "Aside from lateral patterning, the ability to create vertical A-C boundaries with ion beam irradiation could potentially open a way to control buried topological states to yield functionality that is fundamentally different from junctions in traditional electronics."

Reviewer #2 (Remarks to the Author):

In the revised manuscript, the authors have performed a thorough and commendable amount of additional experiments and theoretical calculations to further support their claims and to respond to the reviewers' comments. The authors have responded to all of my previous comments except the request to perform additional experiments with external magnetism to test the Dirac nature of the AC boundary states and their protection by time-reversal symmetry. However, the authors have persuasively stated that these experiments would be complex and beyond the scope of the present work, and have also performed numerical calculations simulating the effects of the external field. I am therefore satisfied with the authors' responses to myself and the other referees, provided that the external magnetic field calculations are included in the final version of the manuscript in either the main text or the Supplementary Material (SM).

We thank the reviewer for their comments. In the revised version, we have added the magnetic-field numerical calculations to the supplementary materials, at the reviewers' suggestion.

Because I previously found this work to rise to the level of impact necessary for publication in Nature Communications, and because I find the revised manuscript to now be clear and compelling – especially with regard to the differences between interfaces and their dimensionality – I recommend this work for publication after the authors address a few final, minor points, which I have enumerated below.

1. In section 4.2 of the SM, the word mapping is misspelled as “mampping.”

Thank you. This has been corrected.

2. In the same section of the SM, the word rectangular is misspelled later in the same paragraph.

Thank you. This has been corrected.

3. In section 6.3 of the SM, the authors briefly refer to the location of the 1D state as a “hinge.” To avoid the ongoing debate as to the nature and presence of 1D “hinge” states in 3D topological insulators. I recommend that the authors change this and all other instances of the word “hinge” to “edge” throughout the main text and SM.

We agree, and have replaced the single instance of “hinge” to “edge” to avoid ambiguity.

Reviewer #3 (Remarks to the Author):

My concerns are fully addressed. Thank you for the additional data and calculations, they are very helpful. In addition, the new calculation results, showing emerging 1D edge channels as surface states are pushed deeper into the bulk, are particularly interesting.

We thank the reviewers for their constructive input into our work, and we aim to write a follow up paper to explore some of these issues in more detail.